# Multi-Concept T2I-Zero: Tweaking Only the Text Embeddings and Nothing Else

## Abstract

Recent advances in text-to-image diffusion models have enabled the photo-realistic generation of images from text prompts. Despite the great progress, existing models still struggle to generate compositional multi-concept images naturally, limiting their ability to visualize human imagination. While several recent works have attempted to address this issue, they either introduce additional training or adopt guidance at inference time. In this work, we consider a more ambitious goal: *natural multi-concept generation using a pre-trained diffusion model, and with almost no extra cost*. To achieve this goal, we identify the limitations in the text embeddings used for the pre-trained text-to-image diffusion models. Specifically, we observe concept dominance and non-localized contribution that severely degrade multi-concept generation performance. We further design a minimal low-cost solution that overcomes the above issues by **tweaking (not re-training)** the text embeddings for more realistic multi-concept text-to-image generation. Our **Correction by Similarities** method tweaks the embedding of concepts by collecting semantic features from most similar tokens to localize the contribution. To avoid mixing features of concepts, we also apply **Cross-Token Non-Maximum Suppression**, which excludes the overlap of contributions from different concepts. Experiments show that our approach outperforms previous methods in text-to-image, image manipulation, and personalization tasks, despite not introducing additional training or inference costs to the diffusion steps.

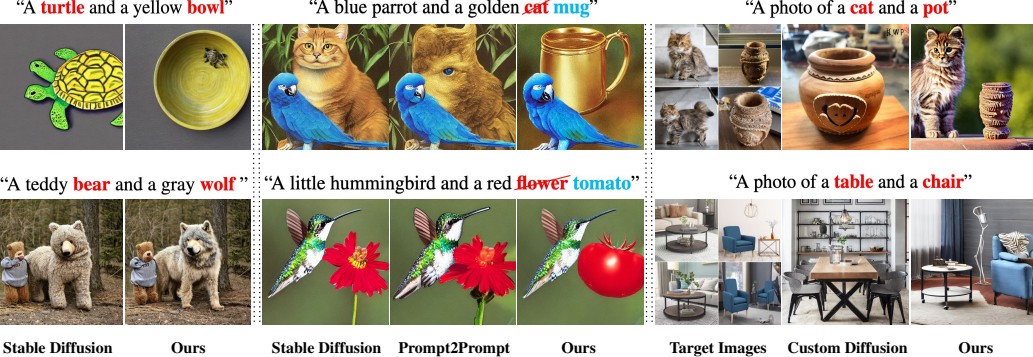

"A **turtle** and a yellow **bowl**"   "A blue parrot and a golden **cat mug**"   "A photo of a **cat** and a **pot**"

"A teddy **bear** and a gray **wolf**"   "A little hummingbird and a red **flower tomato**"   "A photo of a **table** and a **chair**"

Stable Diffusion   Ours   Stable Diffusion   Prompt2Prompt   Ours   Target Images   Custom Diffusion   Ours

Figure 1: **Object Dominance** and **Non-Localized Contribution** make multi-concept image synthesis particularly complex. As a result, existing image manipulation and personalization methods struggle with multiple concepts. Our proposed method addresses these issues and demonstrates its capabilities in **text-to-image**, **image manipulation**, and **personalization** for multiple concepts.

## 1 Introduction

Recently, large-scale diffusion models (Rombach et al., 2022; Saharia et al., 2022) have enabled photo-realistic text-to-image generation. Querying with a text prompt, users receive high-quality images that follow the text description. Numerous works have attempted to further improve the image generation qualities in terms of resolution (Vahdat et al., 2021; Rombach et al., 2022; Saharia et al., 2022), personalization ability (Kumari et al., 2023; Lu et al., 2023), and diversity (Dhariwal & Nichol, 2021; Ho & Salimans, 2022).

Despite the progress, heavy prompt engineering (Xie et al., 2023) is needed to generate visually pleasing images. Moreover, it is hard to obtain high-quality prompts for multi-concept compositional generation since the design space of prompt engineering is greatly enlarged. Users often wish to synthesize specific concepts within the same image, but obtaining ideal multi-concept results is much harder than single-concept ones due to the complexity of text prompts. When generating multiple concepts, it is quite common for the generated image to confuse the attributes of different objects or relations between objects (Kumari et al., 2023; Liu et al., 2022). This is partially due to the limited quality of text prompts in the LAION dataset, where a considerable domain gap exists between the user-generated text prompts and captions of images in the training set (Xie et al., 2023).

Previous works have attempted various fine-tuning methods and utilized inference-time additional guidance. (Kumari et al., 2023; Li et al., 2023; Ma et al., 2023) overcome the limitations by introducing well-designed, high-quality annotations to augment the training dataset. By fine-tuning the diffusion models on images containing multiple concepts and corresponding dense annotations, the models understand multi-concept compositional relationships better. Another line of work (Liu et al., 2022; Chefer et al., 2023; Phung et al., 2023) focuses on improving pre-trained diffusion models with inference-time guidance (Ho & Salimans, 2022). These approaches can sample more realistic multi-concept images by manipulating the conditional distribution through classifier guidance (Dhariwal & Nichol, 2021) and classifier-free guidance (Ho & Salimans, 2022). However, these approaches introduce additional computation costs for each diffusion step at either training or inference time to achieve their desired performance. This prompts the need for a **minimal, low-cost** method approach to generating **multiple concepts**.

We present a novel task of *zero-shot multi-concept text-to-image generation* and outline an economical solution, bypassing the necessity for training or optimizing at each diffusion step. This involves **merely tweaking the text embeddings** employed in text-to-image diffusion models. In our pilot study, we evaluated the efficacy of text embedding and identified two main caveats:

- We pinpoint the first problem of **Concept Dominance**. In certain text prompts, one concept exerts greater influence due to a higher norm, overshadowing other prompts with smaller norms. In such scenarios, the diffusion model is inclined to prioritize the generation of the dominant concept, neglecting the others.

- Even when all concepts are adequately synthesized, we observe a second roadblocker termed **Non-localized Contribution**. Here, the contributions of concepts in the attention maps are distributed across other tokens, including null tokens. This dispersion can be attributed to the pivotal role played by those tokens positioned after the end-of-sentence of the text prompt. Specifically, when these tokens correlate with the semantic regions of a particular concept via cross-attention, it may lead to a distortion of the attended concept in the resultant output.

To address the two-fold challenges, we propose a comprehensive framework. First, we introduce a Corrections-by-Similarities approach to combat the non-localized contribution issue. Specifically, we find similar embeddings and combine them all by their similarity scores to replace the original embedding of a concept. Second, our Cross-Token Non-Maximum Suppression method minimizes the overlap of contributions of different concepts to avoid mixing their features. This technique also helps to retain the contribution of neglected tokens in the first stages of the diffusion process, which leads to the proper generation of all concepts. As shown in Fig. 1, our method not only presents multi-concept text-to-image results more loyal to the text prompts than the original Stable Diffusion, but also delivers more realistic image manipulation and better identity in personalization compared with existing baselines.

Our contributions can be summarized as follows:

- The first pilot study on a minimal, low-cost design of zero-shot multi-concept text-to-image generation. We not only eliminate the need for re-training a pre-trained text-to-image diffusion model, but also ensure that our adjustments solely impact the text embedding prior to the diffusion steps, leaving the diffusion timesteps unaffected.

- Two novel, effective post-hoc techniques to tweak the text embeddings during multi-concept text-to-image generation without re-training. Our Correction by Similarities method tweaks the embedding of concepts by aggregating semantic attributes from most

similar tokens, thereby localizing the contribution. To avoid mixing features of concepts, we further introduce Cross-Token Non-Maximum Suppression, which eliminates overlapping contributions of different concepts.

- Extensive experimental results on multi-concept text-to-image, image manipulation, and personalization demonstrate our superiority against prior methods. Our method effectively addresses the object dominance and non-localized contribution issues, delivering better text-image aligned image synthesis results, more realistic image manipulations, and better identity preservation in personalization for multiple concepts.

## 2 RELATED WORKS

### 2.1 MULTI-CONCEPT TEXT-TO-IMAGE GENERATION

Existing methods for multi-concept text-to-image generation either introduce additional training on multi-concept datasets or require additional optimization-based guidance at each inference step. GLIGEN (Li et al., 2023) proposes gated self-attention layers for multi-concept compositional generation. Custom Diffusion (Kumari et al., 2023) fine-tunes only the attention layers for subject-driven generation. Break-A-Scene (Avrahami et al., 2023) proposes union sampling to enhance the generation of concept combinations. Composable Diffusion (Liu et al., 2022) proposes a framework that composes multiple outputs of a pre-trained diffusion model. StructureDiffusion (Feng et al., 2022) adopts consistency trees or scene graphs to split text prompts into several noun phrases and combine the attention operations afterward. Attend-and-Excite (Chefer et al., 2023) optimizes the cross-attention maps to attend to all subject tokens or excite their activations. Attention-refocusing (Phung et al., 2023) leverages an intermediate spatial layout generated by LLM and further guides the attention maps via the layout predictions. Contrary to the aforementioned methods, our study addresses a particularly challenging low-cost scenario in which we only tweak the text embeddings before starting the diffusion process. This method obviates the need for fine-tuning and incurs no additional costs during each inference diffusion step.

### 2.2 IMAGE MANIPULATION VIA DIFFUSION MODEL

Text-driven image editing has attracted much attention thanks to the remarkable progress in text-driven image synthesis. Early approaches leverage text-guided image inpainting (Lugmayr et al., 2022) or SDEdit (Meng et al., 2021) for simple textural edits. InstructPix2pix (Brooks et al., 2023) constructs a conditional diffusion model based on SDEdit-generated data. Later methods (Hertz et al., 2023; Parmar et al., 2023) study to manipulate the cross-attention maps in the diffusion process for more complex shape edits. Prompt mixing (Patashnik et al., 2023) and Self-guidance (Epstein et al., 2023) further leverage self-attention maps to perform object-level shape control over the generative process. Specifically, in the context of editing multi-concept images, we observe that existing attention-based editing techniques frequently encounter challenges stemming from non-localized contributions inside attention maps. With the implementation of our proposed strategies, optimally formulated text embeddings facilitate superior image editing than existing baselines.

### 2.3 PERSONALIZED TEXT-TO-IMAGE GENERATION

Personalized generation targets generating user-specified concepts and preserving the identity of the concept. DreamBooth (Ruiz et al., 2023) optimizes the whole diffusion model using a special token in the text prompt. Textual Inversion (Gal et al., 2022) optimizes the text embedding while keeping the diffusion model frozen. Custom Diffusion (Kumari et al., 2023) fine-tunes only the attention layers. Perfusion (Tewel et al., 2023), SVDiff (Han et al., 2023), LoRA (Hu et al., 2021) further introduce the decomposition of weight kernels during fine-tuning. Specialist Diffusion (Lu et al., 2023) presents a style-specific personalized model via plug-and-play sample-efficient fine-tuning. SuTI (Chen et al., 2023) adopts apprenticeship learning to mimic the behavior of DreamBooth experts. ELITE (Wei et al., 2023) designs a global and a local mapping network for customization. XTI (Voynov et al., 2023) extends textual inversion into a richer inversion space. NeTI (Alaluf et al., 2023) learns to personalize concepts using a neural representation learned over a space-time conditioning space. Given that our methodology centers on the formulation of suitable text embeddings for multi-concept prompts, we demonstrate that the proposed approach can adaptively

extend the single-concept personalization method, NeTI (Alaluf et al., 2023), to accommodate multi-concept personalization.

# 3 METHOD

**Overview**   In this section, we first discuss the preliminaries of the latent diffusion model and the cross-attention mechanism for text-conditioned image generation. Then, we illustrate in detail the object dominance and non-localized contribution issues observed in the text embeddings for multi-concept text prompts. Finally, we present our proposed low-cost framework, introducing our Corrections-by-Similarities approach and Cross-Token Non-Maximum Suppression method.

## 3.1 PRELIMINARIES

**Latent Diffusion Model**   In this work, we apply our method over Stable Diffusion (Rombach et al., 2022) (SD), which has gained tremendous interest among the community since its release. Unlike Dalle-2 (Ramesh et al., 2022) and Imagen (Saharia et al., 2022) that operate on image space, SD (Rombach et al., 2022) operates on the latent space of a pre-trained variational autoencoder. The training process optimizes the weighted evidence lower bound (ELBO) (Ho et al., 2020; Kingma et al., 2021):

$$\mathcal{L}_{\text{ELBO}}(\theta) = \mathbb{E}\left[w(t)\left\|\epsilon_\theta\left(\alpha_t \boldsymbol{x}_0 + \sigma_t \epsilon; t\right) - \epsilon\right\|_2^2\right], \tag{1}$$

where $\epsilon \sim \mathcal{N}(\mathbf{0}, \mathbf{I})$. $w(t)$ are coefficients found to perform well when set to $w(t) = 1$ (Ho et al., 2020). In the sampling process, we start with $\boldsymbol{x}_T \sim \mathcal{N}(\mathbf{0}, \mathbf{I})$, and then gradually reduce the noise level as we reduce timestep t. Finally, at the end of the iterative process, we reach a clean latent image, which can be further decoded by the pre-trained variational autoencoder into a clean RGB image.

**Cross Attention Mechanism for Text Conditioning**   Text-conditioned image generation is implemented via cross-attention units connecting text embeddings and multi-scale feature maps extracted from the input image. Text embeddings in SD (Rombach et al., 2022) are obtained via a pre-trained CLIP (Radford et al., 2021) text encoder. For an input latent image at resolution $64 \times 64$, the cross-attention layers operate at resolutions of 64, 32, 16, and 8. For an image feature of size $H \times W$ and lengths of text token $N$, we construct cross-attention maps of size $H \times W \times N$, where each token attends to all spatial patches inside the image feature. Specifically, the value at $(i, j, k)$ inside the attention map refers to the information relationship between $k$-th text token and $(i, j)$-th spatial location in the feature map. In this work, we analyze the attention maps and identify the issues brought by text embeddings in multi-concept text-to-image generation.

## 3.2 THE DEVIL IS IN THE TEXT-ENCODER

Modern text-guided image diffusion models, such as LDM (Rombach et al., 2022) and Imagen (Saharia et al., 2022), are designed to generate images with the condition of a sentence. Even though the models are trained on huge datasets, generating multiple concepts still remains challenging. In many cases, the model is not able to generate all concepts properly. The challenges of multi-concept image synthesis can be divided into the following parts: **Object Dominance** - *when some concepts dominate the generation process and others get neglected.* **Non-Localized Contribution** - *when all concepts are synthesized, but their respective embeddings contribution is not localized* (see Fig 4). This behavior limits the opportunities for image manipulations of multiple concepts.

**Object Dominance**   In multi-concept image synthesis, the order of the concepts in the text defines dominance, and the first concept often becomes superior. As can be seen in Fig 2, switching the input embeddings of the text encoder (d), causes switching dominance between concepts. However, when switching output embeddings of the model (c), the dominance is not changed. This is because the order of an embedding in the cross-attention layer is not decisive for dominance. *The text encoder has defined the dominance in one of the embedding vectors beforehand*. Also, reversing the positional encoding of the text encoder (b) does not impose confusion between the dominant and non-dominant concepts. This issue can be solved by tweaking the output of the text encoder before conditioning the diffusion process. We noticed the embedding vector of the dominant concept often

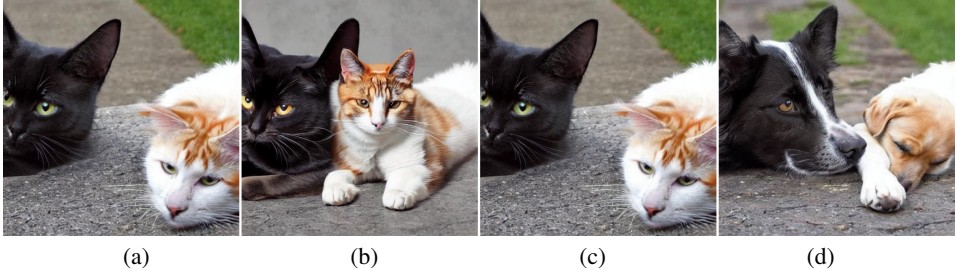

| (a) | (b) | (c) | (d) |

Figure 2: Synthesized images with replacements of **inputs** and **outputs** of text-encoder. *(a) original, (b) reversed positional encoding, (c) switched output embeddings of "cat" and "dog" words, (d) switched input embeddings of "cat" and "dog" words.*

has a higher norm than non-dominant ones. In our simple experiment, we decrease the norm of the dominant concept by a suppression strength $s \in [0, 1]$. A qualitative comparison between the results of **Attend&Excite** (Chefer et al., 2023), **A-STAR** (Agarwal et al., 2023) can be found in Fig 3.

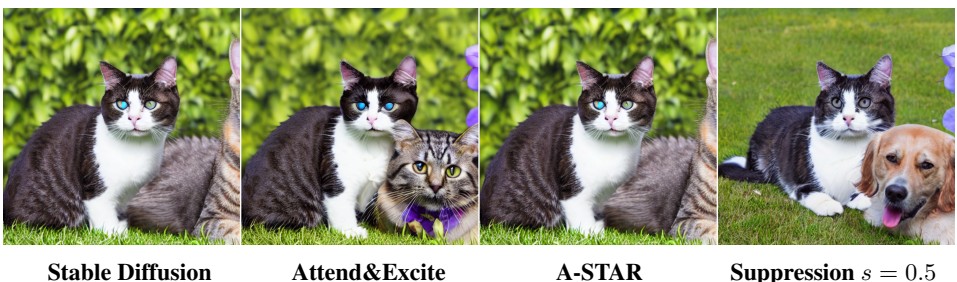

| **Stable Diffusion** | **Attend&Excite** | **A-STAR** | **Suppression** $s = 0.5$ |

Figure 3: A comparison of the results of solutions against concept dominance of *"A photo of a **cat** and a **dog**"* prompt.

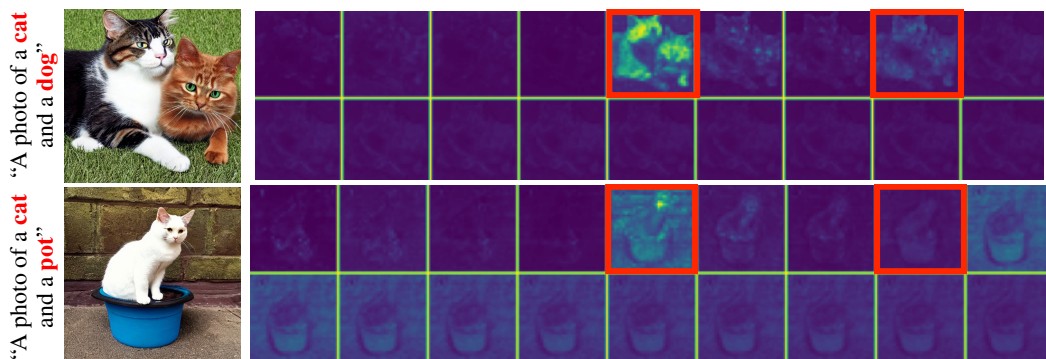

Figure 4: Cross-Attention visualization. In each group, from left to right, the cross-attention maps correspond to the text tokens inside the prompt, followed by padding null tokens. Red tokens are marked with red rectangles. **Top**: *Concept Dominance*, **Bottom**: *Non-Localized Contribution*.

**Non-Localized Contribution**    Multi-concept image manipulation remains a challenging problem. All concepts can be synthesized, but not all of them can be edited. This is due to the non-localized contribution of embedding vectors. A concept can be generated by other embeddings, including the embeddings of null tokens. In the second row of Fig 4, the pot is generated by the contribution of null tokens, and its corresponding attention scores are too low. We believe the text encoder leads to this phenomenon. Often, the embeddings of null tokens contain rich information about the text, such as the features of the scene, main objects, attributes, and others. Due to the nature of the design of text encoders, they often contain more information about the last concept with respect to the order

in the text, than the first one. We computed *cosine similarity* and *Euclidean distance* between the embeddings of the selected concepts and all other embeddings (see Fig 5). It can be noticed that null tokens often are more similar to the non-dominant concept than the dominant one. In the next section, we propose a training-free solution to localize the contribution of concepts.

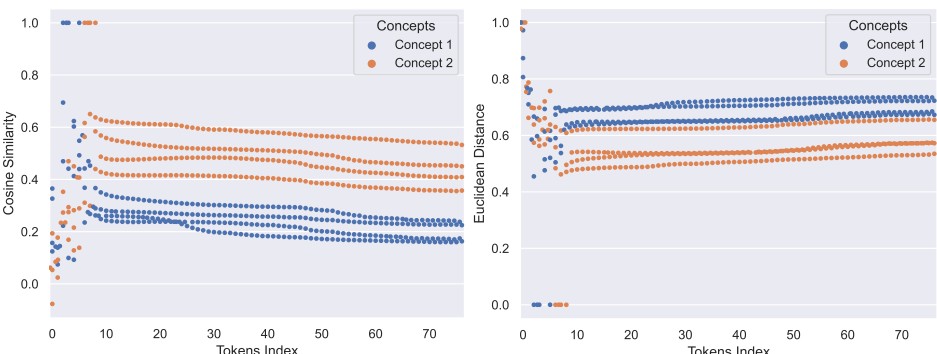

Figure 5: Difference between selected concepts and all other embeddings. All prompts contain less than ten words. *(Left) Cosine Similarity* ↑, *(Right) Normalized Euclidean Distance* ↓.

## 3.3 CONTRIBUTION LOCALIZATION

**Corrections by Similarities** In this section, we propose a zero-shot method, which aims to re-collect features of concepts from most similar embeddings, such as embeddings of null tokens. We tweak the $f \in R^{n \times d}$ output vectors of the text encoder before the cross-attention operation. First, we compute the elementwise product between the embedding of the concept $f_c \in R^d$ and all other embeddings $f_i \in R^d$. Then, we filter the product values that are below a threshold $\tau$ and normalize them to leave only the most similar dimensions between the two vectors. A windowing function is also applied to eliminate the impact on the tokens that are farther away from the selected one in the text. We use these similarity scores to combine all embeddings as a new vector representation.

A new $\tilde{f}_c$ vector will be used to replace the embedding of the selected concept with $\alpha$ blending hyperparameter. This parameter helps to control the strength of the suppression. The $\tilde{f}_c$ vector is considered to be a corrected and recovered version of the previous embedding, which has high attention scores and localized contribution in the respective area of the cross-attention map. The end-to-end algorithm of our proposed method is described in **Algorithm 1**.

---

**Algorithm 1**: Correction by Similarities

---

**1 Input**: Embedding $f \in R^{n \times d}$, selected token indices $C = \{c_1, c_2, ..., c_k\}$, score threshold $\tau \in [0, 1]$, window size $\gamma$, correction strength $\alpha \in [0, 1]$
**2 Output**: Corrected embedding $\tilde{f} \in R^{n \times d}$
**3 for** $c = c_1, c_2, ..., c_k$ **do**
**4**     $S^c \leftarrow f^c \otimes f;$                      $S^c \in R^{n \times d}$
**5**     $\tilde{S}^c \leftarrow \Phi(S^c, \tau)$                $\tilde{S}^c \in R^{n \times d}$
**6**     $\tilde{S}^c \leftarrow \Psi(c, \gamma) \otimes \tilde{S}^c$        $\tilde{S}^c \in R^{n \times d}$
**7**     $\tilde{f}^c \leftarrow \sum_{i=1}^{n} \tilde{S}^c \otimes f$         $\tilde{f}^c \in R^d$
**8**     $\tilde{f}^c \leftarrow (1 - \alpha)f^c + \alpha\tilde{f}^c$    $\tilde{f}^c \in R^d$
**9 end**
**10 Return** $\tilde{f}$

---

where $\otimes$ is the elementwise multiplication operation, $\Phi(S, \tau)$ is a function that thresholds $S$ values with $\tau$ parameter and normalizes them by $\max(S)$, $\Psi(c, \gamma) \in [0, 1]^n$ is a windowing function with window size $\gamma$, which excludes the contribution of tokens farther from the selected token $c$. In the first example of Fig 6, the cross-attention mask of the embedding of the *"pot"* word has low attention scores and the pot was originally generated by the contribution of other tokens. Meanwhile, our method is able to re-scale the contribution by collecting semantic information from the most similar

embeddings, without any training phase. Moreover, our proposed solution recovers the features of neglected concepts and synthesizes all of them, often minimizing the dominance problem (see the second example of Fig 6).

"A photo of a cat and a **pot**"          "A turtle and a yellow **bowl**"

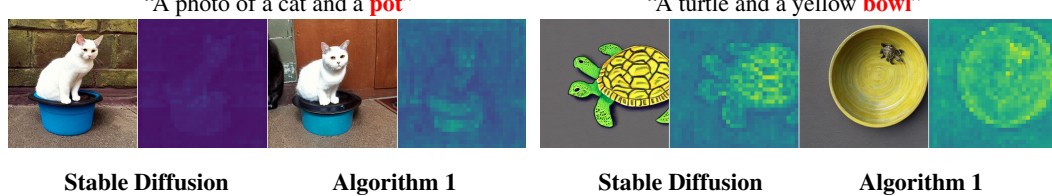

**Stable Diffusion**       **Algorithm 1**        **Stable Diffusion**       **Algorithm 1**

Figure 6: A comparison between **Stable Diffusion** and **Algorithm 1**. Masks are average cross-attention maps of the concept highlighted in **red**.

"A teddy **bear** and gray wolf"          "A blue parrot and a golden **cat**"

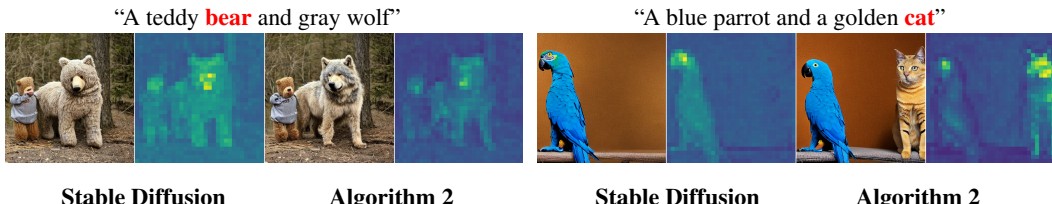

**Stable Diffusion**       **Algorithm 2**        **Stable Diffusion**       **Algorithm 2**

Figure 7: A comparison between **Stable Diffusion** and **Algorithm 2**. Masks are average cross-attention maps of the concept highlighted in **red**.

**Cross-Token Non-Maximum Suppression**   One of the issues that often occur in multi-concept image synthesis is the features mixing of different concepts. The model often generates a hybrid object mixing different features and attributes (see Fig 7). This is due to the overlap of the contribution areas of embedding of corresponding tokens. Motivated by the idea of eDiff-I (Balaji et al., 2022) and the Non-Maximum Suppression algorithm of object detection methods, we propose a method that suppresses weak contributions by using attention maps of the cross-attention layer. The attention map of each token represents the contribution score for every pixel of an image. By leaving the highest contributed token of a pixel, the algorithm excludes the contribution of other tokens by decreasing their attention scores, avoiding mixing the features and attributes of different concepts. A step-by-step implementation of our proposed method can be found in **Algorithm 2**.

---

**Algorithm 2**: Cross-Token Non-Maximum Suppression

---

1 **Input**: Attention Map $A \in R^{n \times h \times w \times t}$, selected token indices $C = \{c_1, c_2, ..., c_k\}$
2 **Output**: Suppressed Attention Map $\tilde{A} \in R^{n \times h \times w \times t}$
3 $A^C \leftarrow A[:, :, :, C]$;    $A^C \in R^{n \times h \times w \times k}$
4 $A^C \leftarrow G(A^C)$;     $A^C \in R^{n \times h \times w \times k}$
5 $M \leftarrow \arg\max_C(A^C)$;  $M \in R^{n \times h \times w}$
6 $\tilde{M} \leftarrow O(M)$;     $\tilde{M} \in R^{n \times h \times w \times t}$
7 $\tilde{A} \leftarrow \tilde{M} \otimes A$;    $\tilde{A} \in R^{n \times h \times w \times t}$
8 **Return** $\tilde{A}$

---

where $G(A)$ is a Gaussian smoothing operator with kernel size $\kappa = 3$ and $\sigma = 1$, $O(M)$ is a suppression function, such as one-hot vector operator with size $t$, $\otimes$ is element-wise product operator.

Our proposed method is able to disentangle mixed features of different concepts, which leads to the proper generation of all concepts. In the first example of Fig 7, attention scores of the "bear" token are high on the contribution area of the "wolf" token, which causes the generation of a hybrid animal. The suppression of low scores in overlapping areas leads to the proper generation of the "wolf". While in the second example of Fig 7, our proposed method helped to synthesize both concepts by retaining high contributions of the "cat" token in the first stages of the diffusion process.

# 4 EXPERIMENTS

## 4.1 MULTI-CONCEPT IMAGE MANIPULATIONS

Modern image manipulation solutions, such as (Hertz et al., 2023; Patashnik et al., 2023), heavily rely on the contribution of word embeddings. Specifically, they use attention maps of selected tokens to do manipulations. Multi-concept image manipulation is another challenging problem because of the **concept dominance** and **non-localized contribution** problems. The embedding vector of the neglected concept does not have a big contribution, thus making the manipulation impracticable. Our designed algorithms (**Algorithm 1**, **Algorithm 2**) improve the contribution localization, which leverages multi-concept image manipulation capabilities. Motivated by the idea of prompt mixing in (Patashnik et al., 2023), we use a simple prompt injection technique in the denoising process. For an original prompt $P_{c_1,c_2,...,c_k}$, with $\{c_1, c_2, ..., c_k\}$ concepts, we replace a concept in the prompt with a new one: $P^*_{c_i \leftarrow \tilde{c}}$ and inject it into the denoising process for timesteps smaller than a predefined $t$ threshold. As shown in Fig 8, our proposed solution expands image editing opportunities for neglected concepts. Moreover, our methods show superiority over other known image manipulation

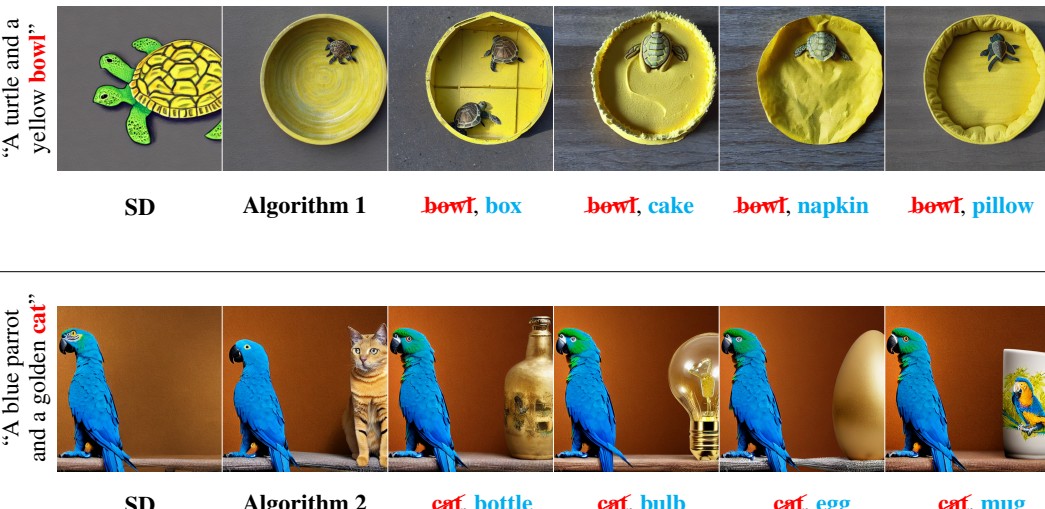

Figure 8: Image manipulation results with our proposed **Algorithm 1** and **Algorithm 2**. The manipulated concept highlighted is in **red**.

techniques such as **Prompt2Prompt** and **Instruct Pix2Pix** (Brooks et al., 2023). We used default values of hyperparameters for both methods. A qualitative comparison can be found in Fig 9.

## 4.2 MULTI-CONCEPT PERSONALIZATION

Personalized text-to-image methods have become very popular in the research community recently. Many solutions, such as **Dreambooth** and **Textual Inversion**, require no more than ten examples of a concept to finetune the diffusion model. However, these models often struggle to work with multiple concepts. Similar to multi-concept image manipulation, the aforementioned issues limit the opportunities for multi-concept personalization. Our solution is applicable to personalization for multiple concepts. Specifically, we adapted the **NeTI** (Alaluf et al., 2023) advanced textual inversion architecture for multiple concepts by extending the number of learnable embeddings for concepts of more than one. To avoid memorizing features of each other, we applied **Supervised Contrastive Loss** (Khosla et al., 2020) between embeddings that are inversed and between their respective outputs of the textual encoder. We used the same prompt injection technique mentioned in Section 4.1. For the first stages of the denoising process, We used general stable diffusion to get the layout and shapes of the image. After layout acquisition, we replace embeddings of concepts with their personalized and inversed vectors for the next stages of the diffusion process. A qualitative comparison can be found in Fig 10.

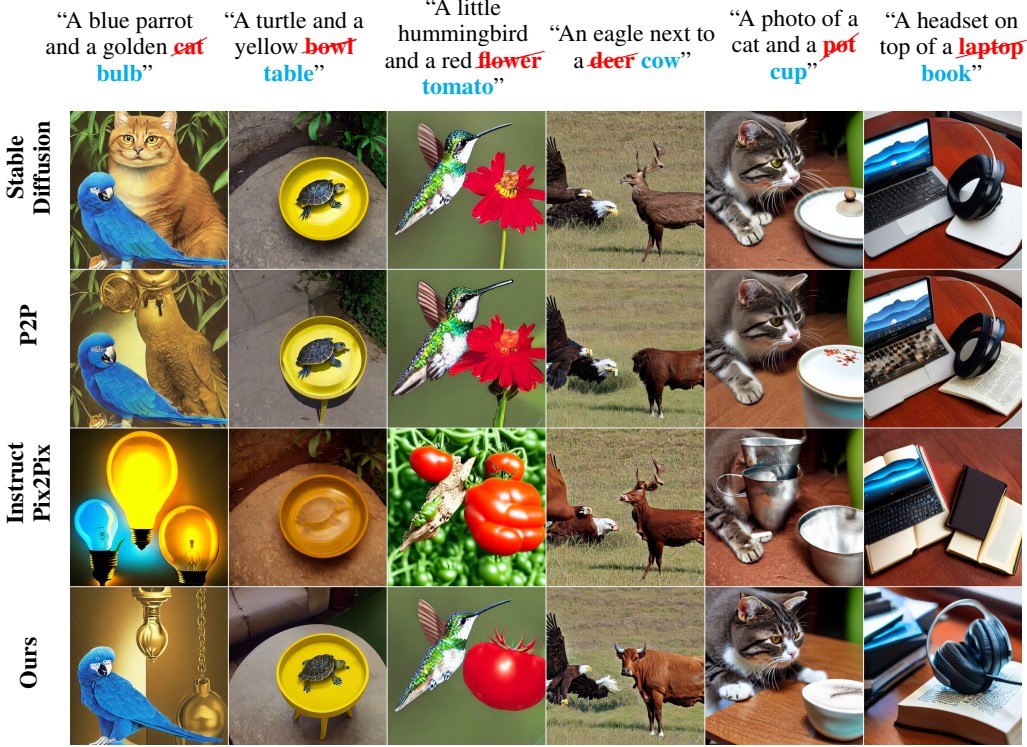

Figure 9: Comparisons on image editing against **Prompt2Prompt** and **Instruct Pix2Pix**.

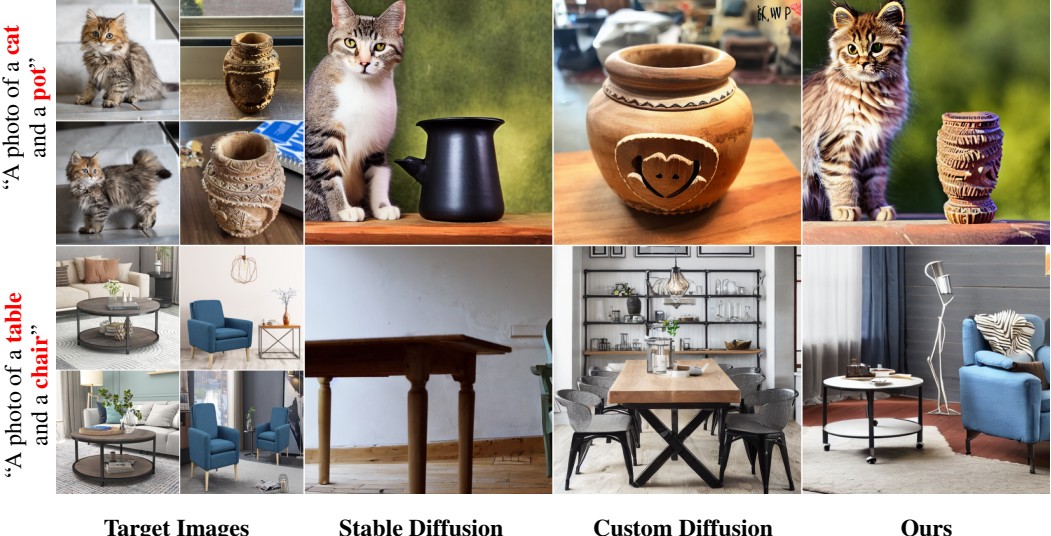

Figure 10: A comparison of personalization results between **Custom Diffusion** and **Ours**.

## 5 CONCLUSION

In this paper, we reach for an ambitious goal: natural multi-concept generation with a pre-trained diffusion model and almost no extra cost. We first identify the concept dominance and non-localized contribution issues in text embeddings. Then, we propose a minimal, low-cost solution that tweaks the text embeddings only without any impact on the diffusion timesteps. With the help of our Corrections-by-Similarities approach and Cross-Token Non-Maximum Suppression method, we outperform existing approaches on text-to-image, image manipulation, and personalization, despite not including extra cost to the diffusion steps.

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
