# OpenReview forum: "Multi-Concept T2I-Zero: Tweaking Only The Text Embeddings And Nothing Else"
_ICLR.cc/2024/Conference — ICLR 2024 Conference Withdrawn Submission_

### Official Review · Reviewer_RDxc · 2023-10-28

**Soundness:** 3 good
**Presentation:** 3 good
**Contribution:** 3 good
**Rating:** 5
**Confidence:** 4

**Summary:**

This paper focus on the multi-concept generation problem of Stable Diffusion, where the the concept dominance and non-localized contribution issues are identified and tackled with corresponding Corrections-by-Similarities approach and Cross-Token Non-Maximum Suppression solutions. The proposed method helps better multi-concept generation in various applications.

**Strengths:**

1. The paper presents some interesting empirical observations in Stable Diffusion multi-subject generation.
2. The proposed method is easy-to-understand and sounds reasonable.
3. The paper is clearly written.

**Weaknesses:**

1. The experiments are a little weak, where only few cases are shown. However, for generative models, different models can generate diverse images with different random seeds. More examples should be given and better with quantitative results to show its effectiveness.
2. There are some hyper-parameters in the algorithms, how could the users find the suitable hyper-parameters?
3. In the customization experiment in Figure 10, it seems the cat is different from the input cat, which may be the side effect of the proposed algorithm.
4. Also, there are many manual designed factors in the proposed method, which may not be so friendly to the users, such as hyper-parameters, assigning concept tokens, etc.

**Questions:**

Please see the weakness part. I am glad to increase the score if my concerns are addressed.

---

### Official Review · Reviewer_9jmE · 2023-10-29

**Soundness:** 2 fair
**Presentation:** 3 good
**Contribution:** 2 fair
**Rating:** 5
**Confidence:** 4

**Summary:**

This paper investigates the multi-concept generation and manipulation in text-to-image diffusion models. Several post-hoc algorithms are proposed to tweak the text embeddings without re-training, which are claimed to be able to improve the multi-concept generation.

**Strengths:**

1. The writing is clear and the overall methods are reasonable.
2. There are adequate qualitative comparisons to demonstrate the effectiveness of the proposed methods on some selected samples.

**Weaknesses:**

1. Lack of quantitative comparisons. It is questionable to compare different methods using only visualizations because they can be cherry-picked. Whether the method is effective is still unknown.
2. Related to the above, if the authors think this is a novel task with no previously used benchmark, they should collect a new dataset of prompts with large diversity and design some evaluation metrics to quantitatively compare their methods and the baseline methods.
3. The ``multi-concept'' seems to be a little bit over-claimed, since all the experiments in this paper consider at most 2 concepts. I would think that the method is not so useful if it cannot support more concepts. To avoid any potential cherry-picking, I suggest that the authors should provide at least 20 examples of image manipulation with more than 4 concepts.

**Questions:**

Please see the weaknesses.

---

### Official Review · Reviewer_U4Xb · 2023-11-05

**Soundness:** 3 good
**Presentation:** 1 poor
**Contribution:** 2 fair
**Rating:** 3
**Confidence:** 5

**Summary:**

This study analyzes the negative effects of conventional CLIP text embeddings in Stable Diffusion on multi-concept image generation. This study points out two phenomena; concept dominance and non-localized contribution. Then, Correction-by-Similarities and Cross-Token Non-Maximum Suppression are proposed to improve the quality of multi-concept image generation. The generated examples show that the proposed methods can improve the image quality, while being applicable to manipulation and personalization on images with multiple concepts.

**Strengths:**

S1. This study aims to solve an important problem on multi-concept image generation of Stable Diffusion, which is an open-sourced text-to-image model based on the CLIP text encoder.

S2. The analysis on concept dominance and non-localized contribution is interesting to understand the reason why Stable Diffusion suffers from the inability to generate multi-concept images.

S3. The two proposed methods are intuitive and make sense.

S4. The experimental results show that the proposed methods are effective especially on multi-concept image manipulation and personalization.

**Weaknesses:**

W1. In-depth analyses and experiments are absent. In addition, there is no quantitative result to compare the proposed methods with previous approaches. Please see the questions below.

W2. The presentation of this paper should be improved. Without any formal definition of each operation in the proposed algorithms, most of the parts are explained in text description. Therefore, I think that the readers cannot understand the details of each operation. For exampl,e in Algorithm 2, how does a Gaussian smoothing operation with kernel size 3 work on 4-dimensional tensor $A^C$?


W3. The proposed approaches cannot be automated for arbitrary text prompts. That is, it seems to require hand-craft engineering for selecting concepts and hyper-parameters in Algorithm 1 and Algorithms 2 for each text prompt.

W4. Although the authors claim to pinpoint two problems, this paper does not include whether the proposed methods directly solve the problems, but just showcase some cases.

W5. Failure cases are not introduced.

**Questions:**

Q1. In Section 4.2, how the supervised contrastive loss is used? Although I understand the details of supervised contrastive loss, there is no detailed explanation how it is applied to multi-concept personalization.

Q2. Can the proposed methods be used for a multi-concept image in terms of generation, manipulation, and personalization, where the number of concepts is larger than 2. How about the number of concepts increase from 2 to 5 or more?

Q3. While the authors employ various hyper-parameters such as score-threshold, window size, correction strength, and the kernel of Gaussian smoothing operator, I wonder how the hyperparameter selection affects the generated images. In addition, are the hyper-parameters different according to text prompts, requiring hand-craft engineering per each text prompt?

Q4. Can each algorithm stand alone without the other algorithm?

Q5. How are the attention maps in this study visualized? Since there are multiple heads in cross-attention layers, explaining the details of visualization would be helpful for better understanding. In addition, the cross-attention map can be different according to the diffusion timesteps during image generation.

Q6. Can the claimed effects be generalized to the multi-head attentions? I wonder whether the different scales between attention heads affect the concept of dominance or not. In addition, regarding cross-token non-maximum suppression, should each pixel be corresponded to one concept token over all attention heads? That is, although Algorithm 2 works for a head of cross-attention, the concept mixing between different attention heads can occur.

Q7. In Algorithm 2, why is the shape of attention map $\mathbb{R}^{n \times h \times w \times w}$ instead of $\mathbb{R}^{n \times h \times w}$?

Q8. In Line 5 of Algorithm 1, why does a threshold use a value between 0 and 1? Since $S^c$ is the result of element-wise multiplication of text embeddings, not a cosine similarity, I think each element of $S^c$ may not be in [0,1].

Q9. I wonder whether the Concept Dominance and Non-Localized Contribution occur together or occur independently according to text prompts.

Q10. I think the number and diversity of examples in this paper is limited to understand the effect of the proposed methods. Can you provide more diverse examples with different length of text prompts and the number of concepts? In addition, using semantically similar multiple concepts for image generation will be an interesting analysis to demonstrate the proposed methods.

Q11. In Figure 2, why are the results of (b) and (d) different? Since CLIP text encoders consist of a stack of attention blocks, the positional embedding determines the location of each token. Did you also switch the positional embedding in attentive pooling of the CLIP text encoder?

Q12. In Figure 6, the authors claim that Algorithm 1 reduces the dominance scale. Can you directly check out the reduced scale to demonstrate the effect of Algorithm 1?